# Characterization of Field-Evolved Resistance to Afidopyropen, a Novel Insecticidal Toxin Developed from Microbial Secondary Metabolites, in *Bemisia tabaci*

**DOI:** 10.3390/toxins14070453

**Published:** 2022-07-01

**Authors:** Ran Wang, Qinghe Zhang, Xuan Zhou, Mi Zhang, Qingyi Yang, Qi Su, Chen Luo

**Affiliations:** 1Institute of Plant Protection, Beijing Academy of Agriculture and Forestry Sciences, Beijing 100097, China; luochen@ipepbaafs.cn; 2Hubei Engineering Technology Center for Pest Forewarning and Management, College of Agriculture, Yangtze University, Jingzhou 434025, China; 202072776@yangtzeu.edu.cn (Q.Z.); 202071658@yangtzeu.edu.cn (X.Z.); 3College of Agriculture and Forestry Technology, Hebei North University, Zhangjiakou 075000, China; zhangmi@163.com (M.Z.); yangqingyi1105@163.com (Q.Y.)

**Keywords:** *Bemisia tabaci*, afidopyropen, insecticidal toxin, cross-resistance, synergism, fitness costs

## Abstract

Afidopyropen, a newly identified chemical, is a derivative of pyripyropene A, which is produced by the filamentous fungus *Penicillium coprobium*. It is a promising novel pesticide applied against whiteflies in agriculture. In this study, the reversion and selection, cross-resistance patterns, synergistic effects, and fitness costs of afidopyropen resistance were studied in a field-developed resistant population of *B. tabaci*. Compared to a reference MED-S strain, the field-developed resistant Haidian (HD) population showed 36.5-fold resistance to afidopyropen. Significant reversion of resistance to afidopyropen was found in the HD population when it was kept with no selective pressure of the insecticide. The HD-Afi strain, developed from the HD population with afidopyropen pressure, developed 104.3-fold resistance to afidopyropen and significant cross-resistance to sulfoxaflor. Piperonyl butoxide (PBO) largely inhibited afidopyropen resistance in the HD-Afi strain, which indicates that P450 monooxygenase could be involved in the resistance. Significant fitness costs associated with afidopyropen resistance were observed in HD-Afi. This study indicates that a rotation of afidopyropen with other chemical control agents could be useful for impeding afidopyropen resistance in *B. tabaci*. In addition, we expanded upon the understanding of resistance to afidopyropen, offering evidence suggesting the importance of devising better strategies for the management of whiteflies.

## 1. Introduction

Afidopyropen is a chemical derivative of the natural product pyripyropene A, which is created by *Aspergillus fumigatus* [1]. It has been classified in accordance with IRAC as the first member of the new pyropene class of the MoA sub-group 9D, which is different from other pyridine azomethine derivatives, including pyrifluquinazon and pymetrozine (MoA sub-group 9B) [2]. Afidopyropen provides powerful effects against notorious sucking and piercing arthropods, such as soybean aphids, pecan aphids, whiteflies, and Asian citrus psyllids, including those that have evolved resistance to other popular chemical agents [3,4,5,6]. Similar to pymetrozine, afidopyropen controls the sucking of insect pests by disturbing their coordination and ability to feed. It acts by overstimulating and eventually silencing vanilloid-type transient receptor potential (TRPV) channels, and it works as a specific modulator of insect TRPV channels against pests [1,2]. Afidopyropen acts swiftly, with rapid feeding cessation, which leads to the decreased transmission of plant viruses [2]. In addition, afidopyropen shows a lack of lethal effects for various orders of non-target insect pests, such as cockroaches, beetles, flies, and mosquitoes [7]. The relative lack of toxicity of afidopyropen to the natural enemies of its target pests indicates that this chemical agent is able to optimize the integration of biological and chemical controls for piercing and sucking insect pests [8]. As a succedaneous insecticide belonging to one novel group of chemical agents, it is believed that afidopyropen can potentially be utilized in rotation with more ordinarily utilized chemical agents to optimize the effects of pest management in target insect populations and, therefore, improve insecticide resistance management programs.

*Bemisia tabaci*, the highly invasive and genetically diverse whitefly, is globally one of the most notorious sucking insects. It harms more than 600 species of plants, both directly, by feeding with its piercing–sucking mouthparts, and indirectly, through the transmission of various plant viruses [9,10]. The management of whiteflies primarily rests upon the use of chemical agents, and over time, moderate to very high resistance to these chemical agents has evolved in field populations of *B. tabaci*, making management of this pest increasingly challenging [11]. Approximately 650 reports of resistance to more than 60 insecticidal agents have been recorded in whiteflies [12], particularly in the last five years, during which resistance to commonly used insecticides, such as cycloxaprid, cyantraniliprole, flupyradifurone, and spirotetramat, has been recorded in different parts of China [13,14,15,16]. Although afidopyropen has been registered in China as an insecticidal compound for its powerful effects against whitefly adults of field populations across China on horticultural crops since 2019, it has been reported that a field-collected population of *B. tabaci* displayed a medium level of afidopyropen resistance [17]. Hence, it could be concluded that field-selection pressure applied by long-term and continual applications of insecticides has likely contributed to the evolution of resistance in this population of whiteflies.

Further studies on the cross-resistance patterns, synergistic effects, and fitness costs of field-developed resistance to chemical agents in insect pests could contribute to the development of strategies for delaying the evolution of resistance [18]. Although a series of publications have been reported on cross-resistance patterns, synergistic effects, and fitness costs of popular chemical agents in whiteflies, there is a lack of published data about whitefly resistance to afidopyropen [19,20,21,22,23]. In this study, after performing reversions and selections with a field-collected afidopyropen-resistant HD population, one strain with a high level of afidopyropen resistance, HD-Afi, and one resistance-reversed strain, HD, were obtained. Then, the HD-Afi strain, HD strain, and reference strain MED-S were utilized to establish the patterns of cross-resistance and assess synergistic effects and fitness costs of afidopyropen resistance in whitefly. Our findings may be valuable for devising strategies of management to delay afidopyropen resistance and sustainably control *B. tabaci*.

## 2. Results

### 2.1. Reversion and Selection of Field-Collected Resistant HD Strain

While the resistance to afidopyropen of F_1_ progeny of the initial field-sampled HD population was measured, the resistance was about 36.5-fold (256.320 mg L^−1^) compared to the reference strain MED-S (LC_50_ = 7.029 mg L^−1^). The levels of resistance to afidopyropen in the unscreened HD and screened HD-Afi strains were measured in each of the generations from F_2_ to F_10_ (Table 1). Selections with afidopyropen of the HD-Afi strain, by using the LC_50_ of each generation, began in the F_2_ progeny of the HD strain, and the levels of resistance to afidopyropen in the HD-Afi strain were measured in every generation from F_2_ to F_10_ to confirm the LC_50_ of each progeny. Resistance in the screened HD-Afi strain enhanced stably from 42.7-fold at F_2_ to 79.8-fold at F_6_ and subsequently stabilized at about 105-fold between F_7_ and F_10_. In the HD strain with no selection, the levels of resistance to afidopyropen declined swiftly at the first six generations (from 36.5- to 5.3-fold) and thereafter stabilized at about 3-fold throughout the subsequent four generations (Table 1).

### 2.2. Cross-Resistance Patterns

The levels of resistance to several popular chemical agents in the screened HD-Afi and unscreened HD strains were compared at F_10_ to establish patterns of cross-resistance (Table 2). Resistance to sulfoxaflor in the HD-Afi strain at F_10_ was 18.9-fold compared to the HD strain at F_10_, indicating that selecting for afidopyropen resistance also resulted in significant cross-resistance to sulfoxaflor. However, little cross-resistance to cyantraniliprole (1.5-fold), flupyradifurone (0.9-fold), imidacloprid (1.1-fold), or thiamethoxam (1.2-fold) was found in the HD-Afi strain (Table 2).

### 2.3. Synergism Tests

Synergistic effects with DEM (diethyl maleate), TPP (triphenyl phosphate), and PBO (piperonyl butoxide) on afidopyropen were determined in the strains of HD-Afi, HD, and MED-S (Table 3). The oxidase inhibitor PBO showed 5.6-fold synergism with afidopyropen in the selected HD-Afi strain but no synergism in either the unselected HD strain at F_10_ or in the MED-S strain, therefore suggesting that oxidative degradation could be associated with afidopyropen resistance. Both the esterase inhibitor TPP and the glutathione depleter DEM indicated little synergistic effects with afidopyropen resistance in the tested strains. It was found that 16.8-fold resistance developed after PBO was utilized, indicating that increased oxidative metabolism is one of the mechanisms resulting in resistance to afidopyropen in the HD-Afi strain.

### 2.4. Fitness Comparisons

Various biological components of fitness were compared between the HD and HD-Afi strains when the resistance ratio of HD-Afi was 31.3-fold higher than that of HD. The development of nymphs and pseudopupae in HD-Afi (16.61 ± 0.28 and 5.14 ± 0.17 days) was not significantly delayed when compared with that in the HD strain (16.31 ± 0.24 and 4.74 ± 0.15 days) (Figure 1A). Moreover, the viability of nymphs and pseudopupae in HD-Afi (63.50 ± 4.52 and 78.62 ± 4.15) was significantly decreased in comparison with that in the HD strain (79.50 ± 3.75 and 83.69 ± 3.19) (Figure 1B). Significantly decreased fecundity was found in HD-Afi (104.48 ± 8.33 eggs/female) compared to that in the HD strain (129.48 ± 7.79 eggs/female) (Figure 2A). Furthermore, the oviposition duration of HD-Afi (11.25 ± 0.79 days) was significantly shorter than that of HD (14.48 ± 0.93 days) (Figure 2B). In the hatching rate comparisons, HD-Afi (87.16 ± 4.49) showed lower egg hatchability than MED-S (89.86 ± 3.68) (Figure 2C). The net reproductive rate of HD-Afi (21.3) was lower than that of HD (35.3), and the relative fitness of HD-Afi was found to be merely 0.60 (Table 4). The above data show that there were fitness costs related to afidopyropen resistance in the HD-Afi strain of *B. tabaci*.

## 3. Discussion

Cases pertaining to the swift reversion of resistance to chemical agents have recently been reported in various field-developed populations of insects with resistance [23,24,25,26]. Afidopyropen acts on the transient receptor potential vanilloid (TRPV) of sucking insects and as a modulator of the Nan-Iav vanilloid TRPV subtype, and then against a series of notorious pests [1,2,3,4,5,6,8]. This research suggests that field-developed, medium-level resistance to afidopyropen was unstable in the HD population of *B. tabaci*, alluding to the ability to exploit this trait in insecticide resistance management tactics. For instance, in reports where high resistance to chlorantraniliprole and abamectin had developed in the field, it was expected that the use of these control agents would be stopped immediately. However, they could be utilized again after some time had passed, and the insect populations could go through withdrawal [24,25]. Recently, it has been reported that one field-collected population of *B. tabaci* reached a medium level of resistance to afidopyropen in China [17]. In such circumstances, if more cases of medium or high afidopyropen resistance are detected in the field, the use of this pesticide needs to be stopped immediately to avoid the development of afidopyropen resistance.

Information about patterns of cross-resistance and synergistic effects in field-sampled populations of pest insects could contribute to the development of sustainable strategies for pest management. Currently, the rotation of insecticides from distinct classes of mode of action with little cross-resistance is one of the popular measures used to lessen the pressure of insecticide selection [11,18]. Previously, we found that the HD population with medium resistance to afidopyropen showed low cross-resistance to sulfoxaflor, while significant cross-resistance to other popular insecticides was not detected [17]. In our present work, after the reversion and selection of a field-sampled resistant HD population, an afidopyropen-selected HD-Afi strain of *B. tabaci* with a high level of resistance was established, and significant cross-resistance to sulfoxaflor without cross-resistance to the other chemical agents was detected. The results of cross-resistance could contribute to the design and implementation of programs of chemical agent rotation to delay the development of resistance to afidopyropen. In addition, elevated metabolic detoxification has been indicated as a crucial mechanism of resistance in various mites and insect pests, and it has been demonstrated that detoxification is firmly related to the constitutive overexpression and induction of cytochrome P450 monooxygenases (P450s) in several populations of *B. tabaci* with resistance to various insecticides [11]. The effects of synergism mediated by inhibitors of metabolic enzymes on the toxicity of insecticides have been extensively studied, and in *B. tabaci*, the three popular inhibitors of metabolic enzymes, namely, piperonyl butoxide (PBO), diethyl maleate (DEM), and triphenyl phosphate (TPP), inhibited various levels of resistance to a series of insecticides [14,16,23,26]. Additionally, we previously observed significant effects of synergism in afidopyropen resistance with the oxidase inhibitor PBO in the field-collected resistant HD strain [17]. Here, with the development of resistance to afidopyropen, we continued to find remarkable synergisms with resistance in the afidopyropen-selected HD-Afi population of *B. tabaci*. It could be concluded that detoxifying enzymes could be among the primary factors contributing to the evolution of afidopyropen resistance similar to other whitefly populations’ resistance to different insecticides [16,17,26]. Furthermore, the synergistic effects of PBO, DEM, and TPP on the toxicity of afidopyropen were partly limited in the HD-Afi strain of *B. tabaci*, indicating that the chosen inhibitors were not all specifically addressing those enzymes involved in afidopyropen metabolism.

Fitness costs are major parameters of biology that are expected to be taken into account when formulating management programs of insect pests and their resistance [27]. Recently, a growing number of cases concerning fitness costs associated with insecticide resistance were reported, and they could be utilized to devise strategies of insecticide resistance [28]. Previously, it has been reported that a variety of insect pests display significant fitness costs related to chemical agent resistance, and most tested resistant individuals have shown prolonged developmental phases, decreased viability, reduced oviposition stage, lower fecundity, and a decreased egg hatching rate [20,29,30,31,32,33]. Our findings show that the afidopyropen-selected HD-Afi population of *B. tabaci* displayed obvious fitness costs compared to the reference HD strain, which showed significantly reversed resistance to afidopyropen. In the HD-Afi strain, obvious disadvantages were not observed in the developmental time period but, rather, in survivability and fecundity when compared with those in the reference HD strain, presenting a relative fitness of 0.60. Hence, it could be concluded that the notable fitness costs can be found in the negative effects on fecundity per female and oviposition stage. More importantly, these biological components of fitness costs related to afidopyropen resistance are expected to be considered as key factors for avoiding the development of resistance, and the above findings may be invaluable in devising management strategies of insecticide resistance [27]. In several previous studies on sucking insect pests, it has been found that fitness costs could lead to the restoration of susceptibility when the use of chemical agents is suspended [30,31,34]. In our present research, the tested strain with afidopyropen resistance had significant reproductive disadvantages in comparison with the reference strain, indicating that the development of resistance to afidopyropen would be postponed in nature. Together with these results, the management strategy of resistance to afidopyropen in whitefly should be formulated with consideration of the associated fitness costs.

## 4. Materials and Methods

### 4.1. Insects

The reference population of *B. tabaci* MED was initially sampled from poinsettia *Euphorbia pulcherrima* grown in Beijing that had not been exposed to insecticides in over ten years [35]. The HD population, with a medium level of afidopyropen resistance, had previously been reported [17]. One strain separated from the HD population’s F_2_ progeny was screened with afidopyropen for nine continuous generations to build the HD-Afi strain, and concentrations of the selections were based on the LC_50_ of each generation of HD-Afi. Another strain of the HD population was reared without exposure to chemical agents to confirm the effect of resistance reversion. All tested populations were reared on plants of cotton (*Gossypium hirsutum* L. var. ‘Shiyuan 321’) at 26 ± 2 °C, a relative humidity of 55 ± 5%, and a photoperiod of 14:10 h light and dark in a growth chamber. For the chemical agent bioassays, as a previous publication describes [15], adults of *B. tabaci* aged up to seven days post-eclosion were collected randomly from the larger populations.

### 4.2. Insecticide and Bioassays

Afidopyropen (catalog# DRE-C10047000) and sulfoxaflor (catalog# DRE-C17015000) were obtained from Dr. Ehrenstorfer (Augsburg, Germany). Cyantraniliprole (catalog# 32372-25MG), flupyradifurone (catalog# 37050-100MG), imidacloprid (catalog# 37894-100MG), thiamethoxam (catalog# 37924-100MG-R), piperonyl butoxide (catalog# 45626-100MG), diethyl maleate (catalog# D97703-100G), triphenyl phosphate (catalog# 241288-50G), dimethyl sulfoxide (catalog# D8418-500ML), and Triton X-100 (catalog# 93443-100ML) were obtained from Sigma Aldrich (Shanghai, China).

### 4.3. Bioassays and Synergism Tests

All of the whitefly bioassays were performed according to the protocols previously described [14]. Specifically, all of the chosen chemical agents were dissolved with dimethyl sulfoxide and diluted for each test concentration using distilled water containing 0.1% Triton X-100. Five concentrations of each chemical agent were used in the bioassays. Twenty-millimeter-diameter cotton leaf discs were dipped in each of the test concentrations of each insecticide for twenty seconds, and four replicates were set up for each of the test concentrations. The cotton leaf discs of bioassays were dried at room temperature and then put into 1.8 mL of agar (15 g L^−1^) in one 60 mm long test tube. Twenty-five to thirty adult whiteflies were introduced to each of the test tubes and then kept in incubators at 26 ± 2 °C, a relative humidity of 55 ± 5%, and a photoperiod of 14:10 h of light and dark, respectively. In each of the bioassays, adult whitefly mortality was checked after 48 h, with motionless whiteflies being considered as dead. The bioassay data were analyzed using the software PoloPlus.

### 4.4. Assessment of Fitness Costs

To assess the fitness costs associated with afidopyropen resistance, two generations of HD and HD-Afi strains were tested according to the published method with minor modifications [23]. Eighteen plants of cotton were equally divided into six individual cages for rearing insects (three for the HD strain and another three for the HD-Afi strain) with three plants in each of the cages. One hundred randomly sampled whitefly adults from each of the two tested strains were put into each one of the cages for a 12 h egg laying stage, and after that, the adult whiteflies were removed from each plant, and ten leaves were chosen from each of the tested cages for further work. After checking each of the chosen leaves by microscope, thirty neonates were kept in a clip cage 3.0 cm in diameter, and the development of the neonates was recorded in each clip cage up until adult whiteflies emerged. Freshly emerged adult whiteflies were introduced onto newly tested leaves to check fecundity until the collected individuals died, and after, that the egg hatching rate produced by the individuals was measured. The net reproductive rate (R_0_) was measured using the formula R_0_ = N_n+1_/N_n_, where N_n_ is number of neonates from their parental generation and N_n+1_ is number of neonates of the offspring. The relative fitness of the HD-Afi afidopyropen-resistant strain was assessed as the ratio of R_0_ of HD-Afi to the R_0_ of the reference HD strain.

### 4.5. Data Analysis

Data analysis of the bioassay results based on different working concentrations of chemical agent was performed using the software PoloPlus (LeOra Software, Berkeley, CA, USA, 2002). In fitness comparisons between the HD and HD-Afi strains, statistical analyses were conducted using the software SPSS (2011). Normality for all the data was checked by performing non-parametric Kolmogorov–Smirnov tests (*p* < 0.05). Data that were normally distributed were analyzed via Student’s *t*-test (*p* < 0.05). Data that were not normally distributed were compared via a non-parametric Mann–Whitney *U*-test (*p* < 0.05).

## Figures and Tables

**Figure 1 toxins-14-00453-f001:**
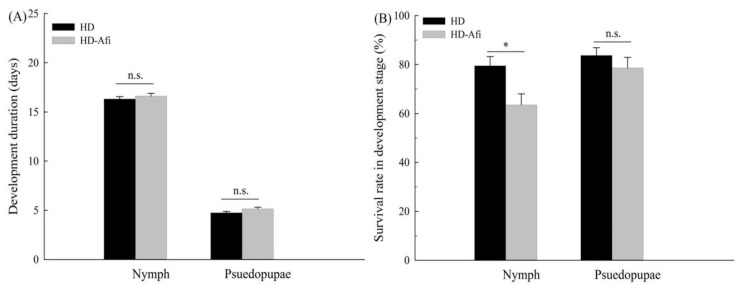
Development duration (**A**) and survival rate (**B**) of nymphs and pseudopupae in HD and HD-Afi strains of *B. tabaci*. Values are presented as the means ± SE. Asterisks above the error bars indicate significant differences (*p* < 0.05), and n.s. indicates not significant (*p* > 0.05).

**Figure 2 toxins-14-00453-f002:**
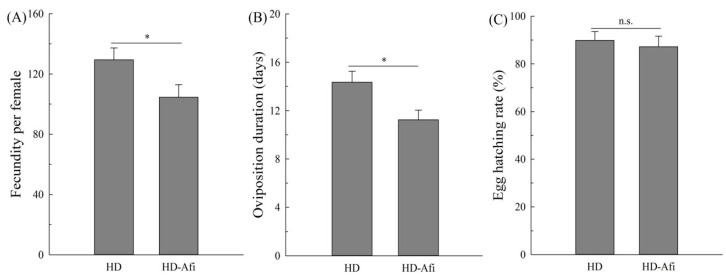
Fecundity (**A**), oviposition duration (**B**), and egg hatching rate (**C**) of HD and HD-Afi strains. Values are presented as the means ± SE. Asterisks above the error bars indicate significant differences (*p* < 0.05), and n.s. indicates not significant (*p* > 0.05).

**Table 1 toxins-14-00453-t001:** Selection of resistance to afidopyropen in the field-collected population of HD.

	HD Strain (without Selection)			HD-Afi Strain (Selected with Afidopyropen)		
G ^a^	LC_50_ (95% CL) ^b^ (mg L^−1^)	Slope ± SE	RR ^c^	LC_50_ (95% CL) ^b^ (mg L^−1^)	Slope ± SE	RR ^c^
1	256.320 (208.229–311.908)	1.315 ± 0.140	36.5			
2	189.807 (137.770–242.631)	1.135 ± 0.139	27.0	299.889 (243.016–362.395)	1.359 ± 0.141	42.7
3	140.582 (108.137–181.141)	1.014 ± 0.133	20.0	380.671 (303.682–458.726)	1.545 ± 0.153	54.2
4	77.052 (58.461–97.009)	1.116 ± 0.137	11.0	476.515 (329.807–624.553)	1.001 ± 0.135	67.8
5	48.763 (33.716–63.451)	1.273 ± 0.153	6.9	534.401 (437.275–634.564)	1.623 ± 0.154	76.0
6	37.446 (31.012–44.505)	1.522 ± 0.144	5.3	560.945 (434.339–691.779)	1.65 ± 0.11	79.8
7	21.139 (14.683–27.455)	1.336 ± 0.152	3.0	740.959 (578.096–922.376)	1.52 ± 0.11	105.4
8	25.320 (20.076–32.049)	1.123 ± 0.137	3.6	715.198 (511.872–914.897)	1.282 ± 0.148	101.7
9	22.536 (18.155–27.733)	1.238 ± 0.137	3.2	762.536 (537.838–984.091)	1.202 ± 0.144	108.5
10	20.470 (17.080–24.573)	1.434 ± 0.139	2.9	733.063 (491.176–970.479)	1.106 ± 0.143	104.3

^a^ Generation of adults used in the bioassay. ^b^ CL = confidence limits. ^c^ RR (resistance ratio) = LC_50_ (selected strain or reversed strain)/LC_50_ (MED-S, 7.029 mg L^−1^).

**Table 2 toxins-14-00453-t002:** Cross-resistance patterns of the afidopyropen-selected HD-Afi strain of *B. tabaci*.

Insecticide	Strain	LC_50_ (mg L^−1^) (95% CL) ^a^	Slope ± SE	RR_1_ ^b^	RR_2_ ^c^
Afidopyropen	MED-S	7.104 (5.836–8.486)	1.465 ± 0.144		
	HD (F_10_)	23.994 (18.802–30.420)	1.076 ± 0.134	3.4	
	HD-Afi (F_10_)	750.818 (603.966–905.157)	1.439 ± 0.145	105.7	31.3
Cyantraniliprole	MED-S	1.040 (0.870–1.232)	1.549 ± 0.146		
	HD (F_10_)	0.883 (0.724–1.069)	1.345 ± 0.139	0.8	
	HD-Afi (F_10_)	1.341 (0.987–1.706)	1.148 ± 0.142	1.3	1.5
Flupyradifurone	MED-S	19.502 (14.602–25.191)	0.989 ± 0.135		
	HD (F_10_)	25.232 (19.527–31.054)	1.340 ± 0.145	1.3	
	HD-Afi (F_10_)	23.825 (19.811–28.979)	1.405 ± 0.141	1.2	0.9
Imidacloprid	MED-S	14.172 (11.067–17.590)	1.193 ± 0.139		
	HD (F_10_)	13.542 (10.896–17.321)	1.176 ± 0.137	1.0	
	HD-Afi (F_10_)	15.078 (11.115–18.976)	1.328 ± 0.149	1.1	1.1
Sulfoxaflor	MED-S	8.860 (6.569–11.148)	1.256 ± 0.144		
	HD (F_10_)	11.033 (8.097–13.925)	1.528 ± 0.143	1.2	
	HD-Afi (F_10_)	208.212 (174.280–243.020)	1.875 ± 0.163	23.5	18.9
Thiamethoxam	MED-S	11.152 (8.187–14.119)	1.220 ± 0.145		
	HD (F_10_)	12.224 (9.812–15.214)	1.192 ± 0.137	1.1	
	HD-Afi (F_10_)	14.291 (9.891–18.734)	0.998 ± 0.136	1.3	1.2

^a^ CL = confidence limits. ^b^ RR_1_ (resistance ratio) = LC_50_ (HD-Afi or HD)/LC_50_ (MED-S). ^c^ RR_2_ (resistance ratio) = LC_50_ (HD-Afi)/LC_50_ (HD).

**Table 3 toxins-14-00453-t003:** Synergism of DEF, DEM, and PBO to afidopyropen in different strains of *B. tabaci*.

Strain	Insecticide/Synergist	LC_50_ (mg L^−1^) (95% CL) ^a^	Slope ± SE	RR	SR ^b^
MED-S	Afidopyropen	7.736 (6.141–9.365)	1.478 ± 0.151		
	Afidopyropen + PBO	7.051 (5.562–9.310)	1.045 ± 0.133		1.1
	Afidopyropen + DEM	7.324 (5.802–9.019)	1.230 ± 0.140		1.1
	Afidopyropen + TPP	7.127 (5.777–9.060)	1.210 ± 0.138		1.1
HD (F_10_)	Afidopyropen	22.002 (16.921–28.872)	0.970 ± 0.132	2.8	
	Afidopyropen + PBO	24.731 (18.825–30.712)	1.275 ± 0.142	3.2	0.9
	Afidopyropen + DEM	25.452 (19.301–33.635)	0.928 ± 0.131	3.3	0.9
	Afidopyropen + TPP	27.871 (22.287–35.190)	1.126 ± 0.134	3.6	0.8
HD-Afi (F_10_)	Afidopyropen	732.112 (575.255–905.446)	1.187 ± 0.135	94.6	
	Afidopyropen + PBO	129.914 (101.106–167.823)	1.016 ± 0.133	16.8	5.6
	Afidopyropen + DEM	754.248 (580.028–1039.142)	0.928 ± 0.131	97.5	1.0
	Afidopyropen + TPP	695.300 (536.987–868.780)	1.129 ± 0.134	89.9	1.1

^a^ CL = confidence limits. ^b^ SR (synergistic ratio) = LC_50_ (afidopyropen only)/LC_50_ (afidopyropen + synergist).

**Table 4 toxins-14-00453-t004:** Fitness component comparisons and relative fitness in field-collected HD strain and laboratory-selected resistant HD-Afi strain of *B. tabaci*.

Fitness Component	HD	HD-Afi
Number of neonates for tests	300	300
Number of pseudopupae	238	190
Number of adults	199	149
Number of female adults	91	70
Mean eggs laid per female	129.5	104.6
Hatchability (%)	89.9	87.2
Predicted neonate number of next generation	10594	6384
Net reproductive rate (R_0_)	35.3	21.3
Relative fitness ^a^	1	0.60

^a^ Relative fitness = R_0_ (HD-Afi)/R_0_ (HD).

## Data Availability

Most of the recorded data are available in all tables in the manuscript.

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
