# Peer review of "Characterization of Field-Evolved Resistance to Afidopyropen, a Novel Insecticidal Toxin Developed from Microbial Secondary Metabolites, in *Bemisia tabaci"

_toxins, 2022, doi:10.3390/toxins14070453_

Round 1
Reviewer 1 Report
The manuscript toxins-1789431 submitted for publication in Toxins is well written and provides interesting data about a novel class of insecticides acting on a receptor present in the chordotonal organ of insects. Its structure is good and the data clearly presented and analyzed.
Nevertheless, I suggest the authors some improvements :
-First, cross-resistance data are provided for several insecticides but the results are not discussed. Such a discussion is lacking and I ask the Authors to modify the manuscript accordingly
-Still about the cross-resistance to sulfoxaflor, I would have liked to know if such resistance to sulfoxaflor is also abolished or reduce by the administration of PBO. If yes, will this reduction be proportional to the one observed for the afidopyropen ?
Actually, these two suggestions are tightly linked and I think that the results dealing with cross-resistance deserve a piece of additional work. The addition of these elements will greatly improve the quality of the manuscript.
-The PBO does not totally abolish the resistance, this result needs to be discussed as no other synergist are able to reduce the resistance level. Does this suggest a modification of the insecticide receptor ?
-I would like the authors to propose a hypothesis in relation to the instability of resistance in the HD population after having maintained it in the laboratory without selection. This reminds me of several papers on similar phenomena, moreover, it has been described with cytochromes P450.
I think that the discussion section deserves more work from the Authors.
I have also less important remarks :
-Place the Table 4 above the figures 1 and 2.
On one hand, afidopyropen is described to be a systemic insecticide and active on sucking and biting insects due to their dietary contamination, on the other hand, I think that the toxicological tests favor a tarsal contamination of the insects. I would like the authors to discuss this apparent contradiction.
I encourage the Authors to submit a modified version of this manuscript according to my remarks, especially those about cross-resistance to sulfoxaflor after a treatment with PBO.
Author Response
Response to Reviewer 1 Comments
The manuscript toxins-1789431 submitted for publication in Toxins is well written and provides interesting data about a novel class of insecticides acting on a receptor present in the chordotonal organ of insects. Its structure is good and the data clearly presented and analyzed.
Nevertheless, I suggest the authors some improvements :
-First, cross-resistance data are provided for several insecticides but the results are not discussed. Such a discussion is lacking and I ask the Authors to modify the manuscript accordingly
Answer: Thanks for the comments and suggestions, and I have added more descriptions on above point in the discussion section.
-Still about the cross-resistance to sulfoxaflor, I would have liked to know if such resistance to sulfoxaflor is also abolished or reduce by the administration of PBO. If yes, will this reduction be proportional to the one observed for the afidopyropen ?
Actually, these two suggestions are tightly linked and I think that the results dealing with cross-resistance deserve a piece of additional work. The addition of these elements will greatly improve the quality of the manuscript.
Answer: Thanks for the comments and suggestions, and in the MS, the cross-resistance to sulfoxaflor in the afidopyropen resistant strain was display with middle level of resistance, almost 20-fold, and we have already established one individual sulfoxaflor-resistant strain from the afidopyropen resistant strain. The sulfoxaflor-resistant strain have been selected with the sulfoxaflor for several generation, and the resistance has reached high level with almost 30-fold. As the reviewer said, we are going to do above work to demonstrate the cross-resistance between afidopyropen and sulfoxaflor.
-The PBO does not totally abolish the resistance, this result needs to be discussed as no other synergist are able to reduce the resistance level. Does this suggest a modification of the insecticide receptor ?
Answer: Thanks for the comments and suggestions, and I have added more descriptions to explain it in the discussion section.
-I would like the authors to propose a hypothesis in relation to the instability of resistance in the HD population after having maintained it in the laboratory without selection. This reminds me of several papers on similar phenomena, moreover, it has been described with cytochromes P450. I think that the discussion section deserves more work from the Authors.
Answer: Thanks for the comments and suggestions, and I have added more descriptions on above points in the discussion section.
I have also less important remarks :
-Place the Table 4 above the figures 1 and 2.
Answer: Thanks for the comments and suggestions, and I have moved the Table 4 above the figures 1 and 2.
On one hand, afidopyropen is described to be a systemic insecticide and active on sucking and biting insects due to their dietary contamination, on the other hand, I think that the toxicological tests favor a tarsal contamination of the insects. I would like the authors to discuss this apparent contradiction. I encourage the Authors to submit a modified version of this manuscript according to my remarks, especially those about cross-resistance to sulfoxaflor after a treatment with PBO.
Answer: Thanks for the comments and suggestions, and firstly, as two other commercial insecticides pyrifluquinazon and pymetrozine, afidopyropen against plant-sucking pests by disturbing their coordination and ability to feed. It acts by overstimulating and eventually silencing vanilloid-type transient receptor potential (TRPV) channels, which consist of two proteins, Nanchung and Inactive, that are co-expressed exclusively in insect chordotonal stretch receptor neurons. In conclusion, afidopyropen acts as a specific modulator of insect TRPV channels for controlling of sucking and biting insects. Secondly, for the the cross-resistance to sulfoxaflor in the afidopyropen resistant strain in the MS, like I mentioned above, after several selections of generations, it has reached higher level of resistance to sulfoxaflor, and we are going to explore it further with not only treatment PBO, but also with digital gene expression via transcriptome sequencing.
Reviewer 2 Report
Reviewers comments#
The manuscript address an interesting issue on resistance obtained from Afidopyropen on the continuous application where they came up with a new idea that the rotation of other chemical control agents can be useful for impeding Afidopyropen resistance in B. tabaci. The manuscript is well written. The work needs certain updates to improve the message and conclusion of the manuscript.
The parameters chosen for strategy building are traditional and informative. But there are certain areas that need more clarification.
Line 16: “P450 monooxygenase could be involved in the resistance” Are there any related proofs or experiences related to this enzyme. Please briefly explain the functional aspect in the discussion section.
How about enzymes related to oxidative stress?
Line 105-106: Why do you select only oxidase and esterase inhibitors (PBO, DEM, and TPP) say something about it. There is no discussion on these inhibitors please provide and give appropriate references?
Line 220: why do you use seven days old adults? Is there any specific cause concerning resistance? Please provide an explanation in M$M Section.
Line 136: How long was this fitness experiment done? (Fig.1) please indicate the duration of the experiment in the Material and Methods section.
Line 140: Fecundity and oviposition got delayed (fig.2 a, b). How do you interpret this data? Please discuss this important variable in the discussion section.
Line 159-161: Suggestion on stopping the application of pesticides is possible. But, how to know when insects acquire resistance at the field level? It is solely dependent on a combination of abiotic and biotic factors. Please provide information on the influence of these factors on acquiring resistance in the pest population.
Line 164-166: Please provide the latest reference (s) supporting the sentence.
Line 181-183: Please provide either the reference for your explanation.
Please provide the mode of action of Afidopyropen? One or two sentences about the mechanism will enrich the discussion part.
General comments:
It can be better to provide the graphical abstract for easy understanding.
Introduction can be improved and elaborated. It is important to include the mechanism of resistance, and synergistic action of the tested pesticides.
The author needs more focus on the recent updates on this present topic please update.
The discussion part lacks supporting references. Some key details related to mechanism and function are missing in the discussion. Please incorporate them for better understanding and elaborate?
Conclusion:
The article provides interesting data but a few parts of the manuscripts are confusing and misleading. Since I found some degree of difficulty in reading and understanding certain parts of the manuscript, the article needs some corrections and clarifications. I do think that the manuscript contains important issues, interesting approaches, and techniques, which can lead in understanding the role of Afidopyropen resistance which can improve the pest control strategies. As suggested, the authors are encouraged to include commented related data for better understanding and to support the conclusion. Therefore, I consider this manuscript suitable for publication after suggested corrections in the Toxins.
Author Response
Response to Reviewer 2 Comments
The manuscript address an interesting issue on resistance obtained from Afidopyropen on the continuous application where they came up with a new idea that the rotation of other chemical control agents can be useful for impeding Afidopyropen resistance in B. tabaci. The manuscript is well written. The work needs certain updates to improve the message and conclusion of the manuscript. The parameters chosen for strategy building are traditional and informative. But there are certain areas that need more clarification.
Line 16: “P450 monooxygenase could be involved in the resistance” Are there any related proofs or experiences related to this enzyme. Please briefly explain the functional aspect in the discussion section. How about enzymes related to oxidative stress?
Answer: Thanks for the comments and suggestions, and I have added more descriptions on above points in the discussion section.
Line 105-106: Why do you select only oxidase and esterase inhibitors (PBO, DEM, and TPP) say something about it. There is no discussion on these inhibitors please provide and give appropriate references?
Answer: Thanks for the comments and suggestions, and I have added more descriptions on inhibitors in the discussion section with the refernces.
Line 220: why do you use seven days old adults? Is there any specific cause concerning resistance? Please provide an explanation in M$M Section.
Answer: Thanks for the comments and suggestions, and using seven days old adults is one common method in the many publications about the bioassays of whitefly adults. I have added more descriptions in the M&M Section with the refernces.
Line 136: How long was this fitness experiment done? (Fig.1) please indicate the duration of the experiment in the Material and Methods section.
Answer: Thanks for the comments and suggestions, and I have added more descriptions in the M&M Section.
Line 140: Fecundity and oviposition got delayed (fig.2 a, b). How do you interpret this data? Please discuss this important variable in the discussion section.
Answer: Thanks for the comments and suggestions, and I have added more descriptions on the results of fecundity and oviposition in the discussion section.
Line 159-161: Suggestion on stopping the application of pesticides is possible. But, how to know when insects acquire resistance at the field level? It is solely dependent on a combination of abiotic and biotic factors. Please provide information on the influence of these factors on acquiring resistance in the pest population.
Answer: Thanks for the comments and suggestions, and I have added more descriptions in the discussion section.
Line 164-166: Please provide the latest reference (s) supporting the sentence.
Answer: Thanks for the comments, and I have added latest references.
Line 181-183: Please provide either the reference for your explanation.
Answer: Thanks for the comments, and I have added latest references.
Please provide the mode of action of Afidopyropen? One or two sentences about the mechanism will enrich the discussion part.
Answer: Thanks for the comments and suggestions, and I have added more descriptions in the discussion section.
General comments:
It can be better to provide the graphical abstract for easy understanding.
Introduction can be improved and elaborated. It is important to include the mechanism of resistance, and synergistic action of the tested pesticides.
The author needs more focus on the recent updates on this present topic please update.
The discussion part lacks supporting references. Some key details related to mechanism and function are missing in the discussion. Please incorporate them for better understanding and elaborate?
Conclusion:
The article provides interesting data but a few parts of the manuscripts are confusing and misleading. Since I found some degree of difficulty in reading and understanding certain parts of the manuscript, the article needs some corrections and clarifications. I do think that the manuscript contains important issues, interesting approaches, and techniques, which can lead in understanding the role of Afidopyropen resistance which can improve the pest control strategies. As suggested, the authors are encouraged to include commented related data for better understanding and to support the conclusion. Therefore, I consider this manuscript suitable for publication after suggested corrections in the Toxins.
Answer: Thanks for the comments and suggestions, as you mentioned above, firstly, I have added one graphical abstract for easy understanding. Then, I have added more descriptions about the mechanism of resistance and synergistic effects on afidopyropen to make the part easy to learn. Besides, I have enriched the discussion part with added references to make the MS clear.
Round 2
Reviewer 1 Report
I think the authors have modified their manuscript "toxins-1789431" enough to make it publishable in Toxins. I would have liked to have a bit more data, especially on the effect of PBO on sulfoxaflor resistance, but I am aware that additional experiments are always difficult to include in a manuscript.
Regarding the incomplete inhibition of afidopyropene resistance in the presence of PBO, I agree that it is possible or even likely that detoxifying enzyme inhibitors are not active on all detoxifying enzymes. Actually, I was also thinking about the modification of the insecticide target as a possible additional mechanism to explain residual resistance.
Sincerely.